# SELF-ADAPTIVE PERTURBATION RADII FOR ADVERSARIAL TRAINING

## ABSTRACT

Adversarial training has been shown to be the most popular and effective technique to protect models from imperceptible adversarial samples. Despite its success, it also accompanies the significant performance degeneration to clean data. To achieve a good performance on both clean and adversarial samples, the main effort is searching for an adaptive perturbation radius for each training sample, which essentially suffers from a conflict between exact searching and computational overhead. To address this conflict, in this paper, firstly we show the superiority of adaptive perturbation radii intuitively and theoretically regarding the accuracy and robustness respectively. Then we propose our novel self-adaptive adjustment framework for perturbation radii without tedious searching. We also discuss this framework on both deep neural networks (DNNs) and kernel support vector machines (SVMs). Finally, extensive experimental results show that our framework can improve not only natural generalization performance but also adversarial robustness. It is also competitive with existing searching strategies in terms of running time.

## 1 INTRODUCTION

The security of machine learning models has long been questioned since most models are vulnerable to perturbations (Papernot et al., 2016). Extremely tiny perturbations may be imperceptible to human beings but yet cause poor performance of models such as deep neural networks (DNNs) (Goodfellow et al., 2014; Madry et al., 2017; Papernot et al., 2017), support vector machines (SVMs) (Xiao et al., 2012; Biggio et al., 2012; 2014) and logistic regression (LR) (Papernot et al., 2016). Examples attacked by such perturbations are generally called as adversarial examples.

To learn robust models, adversarial training has now become one of the most effective and widely-used methods, especially on DNNs and SVMs (Zhou et al., 2012; Kurakin et al., 2017; Miyato et al., 2018; Wang et al., 2019; Shafahi et al., 2019; Wu et al., 2021). However, the success of adversarial training comes at a cost (Tsipras et al., 2018; Zhang et al., 2019). Specifically, as stated in (Tsipras et al., 2018), robustness may be at odds with accuracy, which means models after adversarial training may fail to generalize well on unperturbed examples. It is generally believed that this phenomenon is due to the fixed strength of attack throughout the training process, which ignores the fact that every example may have different intrinsic robustness (Cheng et al., 2020; Zhang et al., 2020).

Naturally, the main effort to mitigate this issue is to find the suitable perturbation radius $\epsilon_i$ for each training sample with explicit or implicit searching strategies. For explicit searching strategies, IAAT (Balaji et al., 2019) uses the brute-force search to find the suitable perturbation radii. MMA (Ding et al., 2018) aims to find the optimal $\epsilon_i^*$ via the bisection search. For implicit searching strategy, Zhang et al. propose an early-stopped PGD strategy called FAT, which adjusts $\epsilon$ implicitly in essence. Although FAT skillfully skips the step of searching $\epsilon_i^*$, it is sensitive to hyperparameters such as steps of PGD attack $\tau$ and the uniform perturbation radius $\epsilon$. Thus, in this paper, we mainly focus on explicit searching strategies. We also give a brief review of the above algorithms in Table 1. From this table, we can see that these searching strategies essentially have an inherent conflict between exact searching and time complexity.

To solve this conflict, in this paper, we propose a novel self-adaptive adjustment framework (SAAT) for perturbation radii. It achieves a better trade-off between natural generalization performance and adversarial robustness without much computational overhead. Firstly, for the adaptive perturbation

Table 1: Comparisons of different adversarial training algorithms which aim at achieving better generalization performance on DNNs and SVMs. (Complexity here refers to the time complexity. $n$ is the training size, $T$ is the number of epochs, $K$ and $\tau$ are the numbers of steps for PGD attack, where $\tau \leq K$, $c_1$ and $c_2$ denote searching steps for $\epsilon_i$.)

| | Algorithm | Problem | Finding suitable $\epsilon_i$ | | Complexity |
| | | | Solving inner problem | Searching strategy | |
|---|---|---|---|---|---|
| DNNs | Standard (Madry et al., 2017) | Minimax | $K$-PGD attack | − | $O(nKT)$ |
| | IAAT (Balaji et al., 2019) | Minimax | $K$-PGD attack | Brute-force search | $O(c_1 nKT)$ |
| | MMA (Ding et al., 2018) | Minimax | $K$-PGD attack | Bisection search | $O(c_2 nKT)$ |
| | FAT (Zhang et al., 2020) | Minimax | $K$-$\tau$-PGD attack | Early-stopped PGD attack | $O(n\tau T)$ |
| | **SAAT-kernel** (Ours) | Minimization | Closed-form solution | Closed-form solution | $O(nT)$ |
| | **SAAT-minimax** (Ours) | Minimax | $K$-PGD attack | Closed-form + fine search | $O(nKT)$ |
| SVMs | adv-SVM (Wu et al., 2021) | Minimization | Closed-form solution | − | $O(nT)$ |
| | **SAAT-SVM** (Ours) | Minimization | Closed-form solution | Closed-form solution | $O(nT)$ |

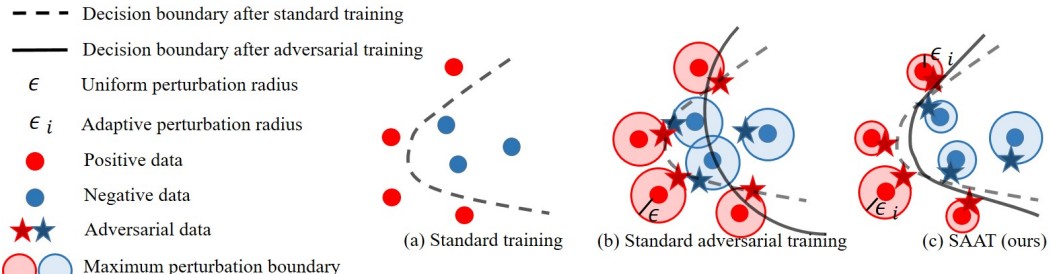

Figure 1: Conceptual illustration of standard adversarial training and our self-adaptive adversarial training (i.e., SAAT).

radius, we intuitively show its superiority in generalization and theoretically illustrate its strength in robustness. Then we design a new learning objective that can construct a self-adaptive perturbation radius for each sample inspired by self-paced learning (SPL) (Jiang et al., 2015). We discuss SAAT on not only DNNs but also kernel SVMs. Correspondingly, we propose two types of optimization algorithms. One is built on the original minimax formulation of adversarial training. It determines the optimal perturbation radii based on the observation that the inner maximization is piecewise approximately linear to perturbation radii. Then we use a fine search to calibrate the values. The other is built on kernel perspective for SVMs and DNNs which transforms the original minimax objective function into an equivalent minimization one. Extensive experimental results show that our framework enjoys better natural generalization performance and higher adversarial robustness compared with other adversarial training algorithms. It is also competitive with existing searching strategies in terms of training time. We summarize the main contributions as follows:

- Theoretically, we prove that adaptive perturbation radii contribute to a lower expected adversarial risk than fixed and uniform perturbation radii, which implies higher robustness against adversarial examples.

- Our self-adaptive adversarial training algorithms can skillfully assign the optimal perturbation radius for each data, which avoids the step of exact searching and achieves a better trade-off between adversarial robustness and natural accuracy.

- The self-adaptive adversarial training strategy that we propose from the kernel perspective is applicable to both SVMs and DNNs. It efficiently optimizes a minimization problem instead of the conventional minimax one since we transform the inner maximization into a simplified and equivalent form.

## 2 PRELIMINARIES

### 2.1 NOTATIONS

We focus on $C$-class classification problems, then the dataset can be defined as $\mathcal{D} = \{x_i, y_i\}_{i=1}^n$, where $x_i \in \mathbb{R}^d$ is the input data, and $y_i \in \{1, \cdots, C\}$ is the label. We will use $\mathbb{I}\{a\}$, the 0-1 loss, to

represent an indicator function, which returns 1 if $a$ is true and 0 otherwise. We will use $l(\cdot)$ to indicate the surrogate loss function of 0-1 loss. The set $\mathcal{B}(\delta, \epsilon) = \{\delta : ||\delta||_p \leq \epsilon\}$ means that the sample is constrained by an $l_p$-normed [1] perturbation $\delta$ with the perturbation radius $\epsilon$. We denote the maximum adversarial loss as $\hat{l}(x, y, f, \epsilon) = \max_{\delta \in \mathcal{B}(\delta, \epsilon)} l(f(x + \delta), y)$, where $f \in \mathcal{F}$ and $\mathcal{F} : \mathcal{X} \to \mathbb{R}$ is one neural network class with depth-$D$ and width-$H$:

$$\mathcal{F} = \{x \to W_D \rho(W_{D-1} \rho(\cdots W_1 x \cdots)), ||W_i||_F \leq M_i, i \in [D]\}. \tag{1}$$

Here $\rho(\cdot)$ is an activation function with $L_\rho$-Lipschitz and $W_i$ is a $H_i \times H_{i-1}$ matrix. Then, we have $H = \max\{H_0, \cdots, H_D\}$, $H_D = 1$ and $H_0 = d$. Thus the class of the maximum adversarial loss can be formulated as $\hat{l}_{\mathcal{F}} = \{\hat{l}_f : f \in \mathcal{F}\}$.

## 2.2 STANDARD ADVERSARIAL TRAINING

The standard adversarial training considers a minimax problem as follows:

$$\min_w \frac{1}{n} \sum_{i=1}^n \max_{\delta_i \in \mathcal{B}(\delta_i, \epsilon)} l(y_i, f_w(x_i + \delta_i)), \tag{2}$$

where $w$ is the model parameter, $x_i + \delta_i$ is the adversarial example of $x_i$. The inner maximization problem actually follows the principle of adversarial attack and aims to construct the most aggressive adversarial examples (Madry et al., 2017), while the outer minimization is to find model parameters to minimize the loss caused by the adversarial examples. It is notable that a fixed and uniform perturbation radius $\epsilon$ is exerted for all training samples here.

## 3 SELF-ADAPTIVE ADVERSARIAL TRAINING

In this section, we first show the superiority of adaptive perturbation radii intuitively and theoretically. Inspired by that, we formulate a novel framework for self-adaptive adversarial training. Then we propose SAAT-minimax to solve the objective.

### 3.1 SUPERIORITY OF ADAPTIVE PERTURBATION RADII

Although adversarial training with adaptive perturbation radii has been widely studied empirically, its theoretical advantages are seldom explored. To fill this vacancy, in the following section, we first intuitively show its superiority on natural generalization and then theoretically illustrate its strength on adversarial robustness.

**Adaptive Perturbation Radii Contribute to Better Generalization Performance.**
Intuitively, as shown in Fig. 1b, for standard adversarial training, the perturbation radii are kept the same for all training samples. However, for samples near the decision boundary, enforcing large perturbation radii will lead to the cross-over mixture of samples in different classes. In this case, it leads to a distorted and undesirable decision boundary and unavoidably destroy the accuracy on unperturbed examples. Thus, we come to the idea of adversarial training with adaptive perturbation radii. As shown in Fig. 1c, the perturbation radii are set according to the specific location of the samples. It effectively avoids the severe distortion of the decision boundary and will not hurt the natural generalization much.

**Adaptive Perturbation Radii Contribute to Lower Adversarial Risk.**
In this part, we theoretically prove that adaptive perturbation radii can lead to a tighter upper bound of adversarial risk than fixed ones in the case of binary classification, which implies higher robustness against adversarial examples.

Firstly, we provide the definition of the expected adversarial risk $\mathcal{R}_{rob}$ as follows:

**Definition 1.** (Expected Adversarial Risk) Following Zhang et al. (2019); Schmidt et al. (2018); Bubeck et al. (2019), to characterize the robustness of a binary classifier $f: \mathbb{R} \to \{0, 1\}$, the expected adversarial risk can be defined as

$$\mathcal{R}_{rob}(f) = \mathbb{E}_{(x,y) \sim \mathcal{D}} \mathbb{I}\{\exists \delta \in \mathcal{B}(\delta, \epsilon) : yf(x + \delta) \leq 0\} \tag{3}$$

---

[1]In this paper, we consider the $l_p$-norm ball of $p \geq 1$ such that the region is convex.

Based on Figure 1 above, we do not want to further increase $\epsilon_i$ if the adversarial example already can be misclassified by the classifier. This leads to the definition of the theoretically optimal adaptive perturbation radius $\epsilon_i^*$ as follows:

**Definition 2.** (Optimal Adaptive Perturbation Radius) Theoretically, the optimal adaptive perturbation radius $\epsilon_i^*$ for each sample can be defined as

$$\epsilon_i^* = \begin{cases} \epsilon_{\max}, & \text{if } \forall \delta_i \in \mathcal{B}(\delta_i, \epsilon_{\max}), \ y_i f(x_i + \delta_i) > 0, \\ \underset{\epsilon_i \leq \epsilon_{\max}}{\arg\min} \ \epsilon_i, \ s.t. \ \exists \delta_i \in \mathcal{B}(\delta_i, \epsilon_i), \ y_i f(x_i + \delta_i) \leq 0, & \text{otherwise.} \end{cases} \quad (4)$$

where $\epsilon_{\max}$ is the maximum perturbation radius, $\epsilon_i$ is the perturbation radius assigned to $x_i$.

*Remark* 3. A few examples of the optimal adaptive perturbation radius can be seen in Figure 1c. For the samples that are misclassified after adversarial attack, $\epsilon_i^*$ is the minimum radii that achieve this goal. For the samples that can be robustly classified even with $\epsilon_{\max}$, $\epsilon_i^*$ equals to $\epsilon_{\max}$.

Before giving our main theorem (i.e., Theorem 5), we provide Assumption 4 as follows.

**Assumption 4.** For the binary classification surrogate loss function $l(\cdot)$, we assume it can be written as $l(f(x), y) = \phi(yf(x))$, where $\phi$ is a non-increasing function and is $L_\phi$-Lipschitz.

Examples of satisfied loss functions include hinge loss, logistic loss(Xiang, 2011), exponential loss (Wyner, 2003) and many others. Based on Assumption 4, the upper bound to the expected adversarial risk can be gotten as follows, the detailed proof is in the appendix.

**Theorem 5.** *When Assumption 4 holds, for any $\omega \in (0,1)$ and any $\hat{l}_f \in \hat{l}_\mathcal{F}$, with probability at least $1 - \omega$, the following holds:*

$$R_{rob}(f) \leq \frac{1}{n} \sum_{i=1}^n \hat{l}(x_i, y_i, f, \epsilon_i^*) + 3B\sqrt{\frac{\log 2/\omega}{2n}} + \frac{24B}{\sqrt{n}} L_\phi L_\rho^{D-1} \max\{1, d^{\frac{1}{2}-\frac{1}{p}}\}(X_p + \epsilon_{\max})Q.$$

*where* $X_p = \max\{\|x_i\|_p\}_{i=1}^n$, $Q = \frac{24B}{\sqrt{n}} L_\phi L_\rho^{D-1} \max\{1, d^{\frac{1}{2}-\frac{1}{p}}\}(X_p + \epsilon_{\max})\sqrt{\log \prod_{i=1}^D \frac{\pi^{H_i H_{i-1}/2}}{\Gamma(\frac{H_i H_{i-1}}{2}+1)} M_i^{H_i H_{i-1}}} \prod_{i=1}^D M_i$ *and* $\Gamma$ *means the gamma function.*

Then we give Theorem 6 to show that the maximum loss function $\hat{l}(x_i, y_i, f, \epsilon)$ increases with regard to $\epsilon$. The detailed proof is provided in the appendix.

**Theorem 6.** *The maximum loss function $\hat{l}(x_i, y_i, f, \epsilon)$ is an increasing function with regard to the perturbation radius $\epsilon$.*

*Remark* 7. Combing Theorem 5 with Theorem 6, it is evident that replacing $\epsilon_{\max}$ with $\epsilon_i^*$ will contribute to a tighter upper bound for the expected adversarial risk $\mathcal{R}_{rob}$. It indicates that adaptive perturbation radius in training stage is a better choice than fixed and uniform radius that can lead to higher adversarial robustness.

## 3.2 FRAMEWORK OF SELF-ADAPTIVE ADVERSARIAL TRAINING

Although several methods have been proposed to search for a suitable perturbation radius for each training sample, there exists a conflict between exact searching and computational load, as mentioned in Section 1 and Table 1. To achieve fast self-adaptive adversarial training, we creatively introduce a self-adaptive regularizer of perturbation radii (i.e., $-\lambda \frac{1}{n} \sum_{i=1}^n \epsilon_i$) into formulation (2), and give our new formulation of self-adaptive adversarial training (SAAT) as follows:

$$\min_{w,\epsilon} \frac{1}{n} \sum_{i=1}^n \left\{ \max_{\|\delta_i\|_p \leq \epsilon_i} l(y_i, f_w(x_i + \delta_i)) - \lambda \epsilon_i \right\}, \quad (5)$$

$$s.t. \ \epsilon_i \in [0, \epsilon_{\max}], \ i = 1, \dots, n.$$

where $\epsilon_i$ is the customized perturbation radius of $x_i$ achieved by the self-adaptive item and $\lambda$ is the regularization parameter. Thus $\epsilon_i$ can update dynamically as the maximum adversarial loss of $x_i$ changes.

*Remark* 8. Note that a similar term is used in self-paced learning (SPL) (Jiang et al., 2015). The core idea of SPL is to learn a model by gradually including samples from easy to complex according to their losses since SPL decides whether the samples can be selected into training via a self-paced regularization. Inspired by SPL, we aim to assign a specific perturbation radius $\epsilon_i$ to each sample according to its loss. Formally, we design a self-adaptive regularizer imposed on $\epsilon_i$ and add it to the original formulation of adversarial training.

### 3.3 SAAT-MINIMAX

In this part, we aim to optimize the SAAT framework (5). Specifically, we propose a two-stage search strategy to find the approximate optimal perturbation radii. The first stage is built on the closed-form solution via the piecewise approximate linearity of $\hat{l}(x_i, y_i, f_w, \epsilon_i)$ wrt. the perturbation radii $\epsilon_i$. The second stage is a fine search to calibrate the results of the first stage. In the following, we will discuss the two-stage search strategy in detail.

Firstly, we observe that $\hat{l}(x_i, y_i, f_w, \epsilon_i)$ is piecewise approximately linear with regard to $\epsilon_i$ for each sample as shown in Fig. 2 and propose Assumption 9. This assumption is verified in the appendix.

**Assumption 9.** $\hat{l}(x_i, y, f_w, \epsilon_i)$ is piecewise linear with regard to $\epsilon_i$ as follows:

$$\hat{l}(x_i, y_i, f_w, \epsilon_i) = \max(0, k\epsilon_i + b) \quad (6)$$

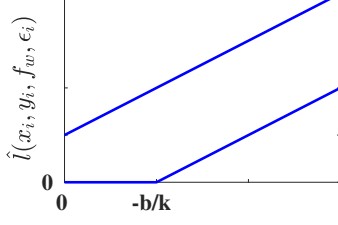

Figure 2: The sketch map of $\hat{l}_f(z_i, \epsilon_i)$ wrt. $\epsilon_i$.

where $k > 0$ is the slope of $\hat{l}$ with regard to $\epsilon_i$ and $b$ denotes $y$-intercept.

Then, with the aid of Assumption 9, we come to Theorem 10, which provides the approximate optimal perturbation radii $\epsilon_i^*$ for optimizing objective function (5). Its proof is presented in the appendix. The detailed setting of $k_i$ and $b_i$ can be seen in section 5.1.3.

**Theorem 10.** *For the minimization problem* $\min_{\epsilon_i \in [0, \epsilon_{max}]} \hat{l}(x_i, y_i, f_w, \epsilon_i) - \lambda\epsilon_i$, *if $f_w$ is given, and Assumption 9 holds, we have the optimal $\epsilon_i^*$ as follows:*

$$\epsilon_i^* = \begin{cases} 0, & \text{if } b_i \geq 0 \text{ and } k_i \geq \lambda; \\ -\frac{b_i}{k_i}, & \text{if } b_i < 0 \text{ and } k_i \geq \lambda; \\ \epsilon_{max}, & \text{otherwise.} \end{cases} \quad (7)$$

The second stage is a fine search to calibrate the results of Theorem 10. Since Assumption 9 may not hold exactly, we use a simple search strategy with a fixed step size to find more accurate values of $\epsilon_i^*$. Specifically, if the PGD attack fails to find an adversarial image $x_i + \delta_i$ that can be misclassified, it implies $\epsilon_i^*$ is too small. Thus, we set $\epsilon_i^* = \epsilon_i^* + \eta$. Otherwise, we set $\epsilon_i^* = \epsilon_i^* - \eta$, where $\eta$ is a pre-specified fixed step size.

Finally, we combine the above two-stage search strategy with the standard adversarial training procedure and give the pseudo-code of our SAAT-minimax in Algorithm 1.

## 4 SELF-ADAPTIVE ADVERSARIAL TRAINING FROM KERNEL PERSPECTIVE

As we all know, traditional adversarial training aims to optimize a minimax problem. It typically uses a gradient-based iterative solver such as multi-step PGD to approximately solve the inner problem, which often leads to high computational overhead. To solve this problem, we propose a new self-adaptive adversarial training strategy from kernel perspective[2]. Specifically, it efficiently transforms the minimax problem (5) into an equivalent minimization one. Then we discuss the detailed self-adaptive adversarial training algorithms via the kernel perspective for both DNNs and SVMs.

---

[2]The kernel perspective means our function $f$ is in the reproducing kernel Hilbert space (RKHS) (Iii, 2004).

---

**Algorithm 1** SAAT-minimax with $l_\infty$-norm constrained perturbations

---

**Input:** $\mathcal{D}$ : training set; $T$ : number of epochs; $\epsilon_{max}$ : maximum perturbation radius; $\gamma$ : learning rate; $K$ : PGD steps; $\alpha$ : PGD step size; $B$ : batch size.

**Output:** $w$.

 1: **for** epoch= $1, \cdots, T$ **do**
 2:     Choose a batch of training samples $\{(x_i, y_i)\}_{i=1}^B \sim \mathcal{D}$.
 3:     Obtain $\epsilon_i^*$ via Theorem 10.
 4:     $\epsilon_i^* = \max(\min(\epsilon_i^*, \epsilon_{max}), 0)$.
 5:     **for** $k = 1, \cdots, K$ **do**
 6:         $\delta_i = \delta_i + \alpha \cdot \text{sign}(\nabla_{\delta_i} l(y_i, f_w(x_i + \delta_i)))$.
 7:         $\delta_i = \max(\min(\delta_i, \epsilon_i^*), -\epsilon_i^*)$.
 8:         Calibrate $\epsilon_i^*$ via the fine search strategy.
 9:     **end for**
10:     $w = w - \gamma \nabla_w l(y_i, f_w(x_i + \delta_i))$.
11: **end for**

---

### 4.1 PRIMARY RESULTS FROM KERNEL PERSPECTIVE

The transformation of the minimax problem (5) contains two steps: firstly we map the perturbations from linear to kernel spaces, then we can solve the unconstrained equivalent form of the inner minimization.

We first discuss the kernelization of the perturbations $\delta$. For an adversarial example $x + \delta$ in the linear space, it is known that if we map it into the kernel space, the kernelized example $\phi(x + \delta)$ will be unpredictable, here $\phi(\cdot)$ is the feature mapping function. Fortunately, Theorem 14 in (Xu et al., 2009) provides a tight connection between perturbations in the linear and kernel space, i.e., the perturbation range of $\phi(x) + \delta_\phi$ tightly covers that of $\phi(x + \delta)$, where $\delta_\phi$ is the perturbation in the kernel space and $\|\delta_\phi\|_2 \leq \sqrt{2f(0) - 2f(\epsilon)}$. Since we can use a $l_2$-norm ball to wrap a $l_p$-norm ball, e.g., $\{\|\delta\|_\infty \leq \epsilon\} \subseteq \{\|\delta\|_2 \leq \sqrt{2}\epsilon\}$, this theorem is applicable to other norms as well.

Based on it, our formulation of self-adaptive adversarial training (5) can be rewritten as the following form in the RKHS $\mathcal{H}$:

$$\min_{f \in \mathcal{H}, \epsilon'} \frac{1}{n} \sum_{i=1}^n \left\{ \max_{\|\delta_\phi^i\|_2 \leq \epsilon_i'} l\left(y_i, \langle f, \phi(x_i) + \delta_\phi^i \rangle_{\mathcal{H}}\right) - \lambda \epsilon_i' \right\}, \tag{8}$$
$$s.t. \ \epsilon_i' \in [0, \epsilon_{\max}'], \ i = 1, \ldots, n.$$

where $\epsilon_i' = \sqrt{2f(0) - 2f(\epsilon_i)}, \epsilon_{\max}' = \sqrt{2f(0) - 2f(\epsilon_{\max})}$.

Then we can obtain the simplified and equivalent form of the inner maximization of Eq. (8) via Theorem 11. The detailed proof can be found in the appendix.

**Theorem 11.** *If $f$ is a function in an RKHS $\mathcal{H}$, the inner maximization problem $\max_{\|\delta_\phi^i\|_2 \leq \epsilon'} l(y_i, \langle f, \phi(x_i) + \delta_\phi^i \rangle_{\mathcal{H}})$ in (8) is equivalent to the regularized loss function $l\left(y_i, f(x_i) + \epsilon' \|f\|_{\mathcal{H}}\right)$, where $\|\cdot\|_{\mathcal{H}}$ stands for the norm in the RKHS.*

According to this theorem, our goal turns to optimize the following minimization problem:

$$\min_{f \in \mathcal{H}, \epsilon'} \frac{1}{n} \sum_{i=1}^n \left\{ l\left(y_i, f(x_i) + \epsilon_i' \|f\|_{\mathcal{H}}\right) - \lambda \epsilon_i' \right\}. \tag{9}$$
$$s.t. \ \epsilon_i' \in [0, \epsilon_{\max}'], \ i = 1, \ldots, n.$$

For the new problem (9), it is obvious that Theorem 10 can be easily applied here to get the optimal perturbation radius $\epsilon_i'^*$ as well, since we denote $l\left(y_i, f(x_i) + \epsilon_i' \|f\|_{\mathcal{H}}\right)$ as $\hat{l}(x_i, y_i, f, \epsilon_i')$.

In this case, we give the optimization framework of SAAT from the kernel perspective in Algorithm 2, which clearly shows the alternative updating for $\{\epsilon_i'^*\}_{i=1}^n$ and function $f$. In the following subsection, we will discuss its applications on DNNs and kernel SVMs in detail.

---

**Algorithm 2** SAAT on Kernel Perspective

---

**Input:** $\epsilon'_{max}, \lambda_0, \mu$.
**Output:** $\{\epsilon'^*_i\}_{i=1}^n$.
 1: Initialize $\lambda = \lambda_0$.
 2: **while** not converged **do**
 3:     Update $\{\epsilon'^*_i\}_{i=1}^n$ via Theorem 10 with fixed $f$.
 4:     Update $f$ with fixed $\{\epsilon'^*_i\}_{i=1}^n$ on DNNs or kernel SVMs.
 5:     $\lambda \leftarrow \mu\lambda$.
 6: **end while**

---

## 4.2 SPECIFIC ALGORITHMS ON DNNs AND SVMs

### 4.2.1 SAAT-KERNEL ON DNNs

Since the RKHS norm $\|f\|_{\mathcal{H}}$ cannot be computed on DNNs, we use the lower bound of $\|f\|_{\mathcal{H}}$ proposed in (Bietti et al., 2019) to approximate its value:

$$\|f\|_{\mathcal{H}} \geq \|f\|_{\delta}^2 := \sup_{\|\delta\|_2 \leq 1} f(x+\delta) - f(x). \tag{10}$$

In this way, since the optimal solution for the perturbation radii $\epsilon'_i$ has already been attained, we can easily optimize learning objective (9) via optimization algorithms such as SGD (Bottou, 2010) and ADAM (Kingma & Ba, 2014). The procedures to alternatively optimize $\{\epsilon'^*_i\}_{i=1}^n$ and the model function $f$ is shown in Algorithm 2.

### 4.2.2 SAAT-SVM ON KERNEL SVMs

Similar with SAAT on DNNs, SAAT on kernel SVMs can be formulated as the following problem:

$$\min_{f \in \mathcal{H}, \epsilon'} \frac{\|f\|_{\mathcal{H}}^2}{2} + \frac{C}{n}\sum_{i=1}^n \left\{ l(y_i, f(x_i) + \epsilon'_i\|f\|_{\mathcal{H}}) - \lambda\epsilon'_i \right\}. \tag{11}$$

$$s.t. \ \epsilon'_i \in [0, \epsilon'_{\max}], \ i = 1, \ldots, n.$$

where $\frac{1}{2}\|f\|_{\mathcal{H}}^2$ is the added norm similar to the SVM formulation in (Dai et al., 2014). As the doubly stochastic gradient descent (DSG) algorithm (Dai et al., 2014) has been proved to be a powerful technique for scalable kernel learning, here we use it optimize Eq. (11). The detailed optimization procedure is provided in the appendix.

## 5 EXPERIMENTS

In this section, we compare SAAT with different adversarial training algorithms on MNIST (Lecun & Bottou, 1998), CIFAR10 and CIFAR100 (Krizhevsky & Hinton, 2009) under $l_2/l_\infty$-norm constrained perturbations on DNNs. Due to the page limit, we only show partial results of $l_\infty$ norm in the following, other results are presented in the appendix. Experiments on kernel SVMs and the verification of Assumption 9 are also presented in the appendix.

### 5.1 EXPERIMENTAL SETUP

#### 5.1.1 COMPARED ALGORITHMS:

- **Natural**: Natural model training on DNNs which minimizes the cross entropy loss.
- **Standard** (Madry et al., 2017): The standard adversarial training method which uses the $K$-step PGD as an attacker.
- **IAAT** (Balaji et al., 2019): Instance adaptive adversarial training which uses brute-force search to assign instance-specific perturbation radius $\epsilon_i$ to each sample.
- **MMA** (Ding et al., 2018): Max-margin adversarial training which directly maximizes the distances from inputs to the decision boundary via binary search for the optimal perturbation radii.
- **FAT** (Zhang et al., 2020): A friendly adversarial training strategy which generates friendly adversarial data by stopping the adversarial data searching algorithms early.

- **TRADES** (Zhang et al., 2019): This method aims to achieve a trade-off between robustness and accuracy via decomposing the robust error as the sum of natural error and boundary error.

- **SAAT-kernel**: Our self-adaptive adversarial training algorithm on DNNs from the kernel perspective. We apply both the hinge loss and the cross entropy loss in the experiments, i.e., SAAT-kernel$_h$ and SAAT-kernel$_c$.

- **SAAT-minimax**: Our self-adaptive adversarial training algorithm on the minimax problem for DNNs. We apply both the hinge loss and the cross entropy loss in the experiments, i.e., SAAT-minimax$_h$ and SAAT-minimax$_c$.

Table 2: Test accuracy (%) of various defense methods trained on MNIST with $l_\infty$-norm constrained perturbations on DNNs. (The results of Natural on clean data are just baselines for reference.)

| Method | Clean | FGSM | 10-PGD | CW | AutoAttack |
|---|---|---|---|---|---|
| Natural | *98.83±0.29* | 14.84±0.66 | 0.00±0.00 | 0.00±0.00 | 0.00±0.00 |
| Standard | 97.82±0.35 | 95.00±0.78 | 89.55±0.89 | 87.27±0.44 | 86.14±0.81 |
| IAAT | 98.57±0.39 | 92.54±0.67 | 85.17±1.04 | 83.64±0.79 | 79.13±0.64 |
| MMA | 98.85±0.63 | 92.87±1.07 | 81.87±0.88 | 75.33±1.19 | 70.32±0.94 |
| FAT | 97.08±0.59 | 94.11±0.94 | 80.47±0.86 | 78.92±0.77 | 66.53±0.65 |
| TRADES | 98.78±0.47 | 97.07±0.52 | 90.44±0.66 | 86.74±0.67 | 88.09±0.46 |
| SAAT-kernel$_h$ | 98.47±0.33 | 78.92±0.72 | 67.76±0.85 | 62.45±0.46 | 52.79±0.46 |
| SAAT-kernel$_c$ | 98.61±0.29 | 80.55±0.68 | 62.86±0.55 | 63.39±0.74 | 53.64±0.67 |
| SAAT-minimax$_h$ | 98.44±0.67 | **97.42±0.82** | **94.80±0.76** | **94.42±0.59** | **93.35±0.46** |
| SAAT-minimax$_c$ | **98.88±0.27** | 96.29±0.54 | 91.97±0.69 | 92.27±0.74 | 90.03±0.57 |

Table 3: Test accuracy (%) of various defense methods trained on CIFAR10 with $l_\infty$-norm constrained perturbations on DNNs. (The results of Natural on clean data are just baselines for reference.)

| Method | Clean | FGSM | 10-PGD | CW | AutoAttack |
|---|---|---|---|---|---|
| Natural | *92.26±0.45* | 5.79±0.36 | 0.00±0.00 | 0.00±0.00 | 0.00±0.00 |
| Standard | 75.65±0.37 | 62.73±1.09 | 48.51±0.64 | 40.37±0.84 | 42.44±0.63 |
| IAAT | 74.56±0.52 | 49.88±0.82 | 38.02±0.69 | 36.79±0.74 | 30.66±0.79 |
| MMA | 79.15±0.67 | 62.83±0.74 | 45.73±0.82 | 44.34±0.96 | 34.28±0.68 |
| FAT | 86.80±0.45 | 61.49±0.62 | 46.17±0.77 | 45.14±0.83 | 33.34±0.87 |
| TRADES | 86.74±1.27 | 63.42±0.64 | 49.76±0.65 | 45.39±0.76 | 49.55±0.72 |
| SAAT-kernel$_h$ | 83.89±0.37 | 24.97±0.79 | 18.54±0.82 | 16.33±0.87 | 11.95±0.58 |
| SAAT-kernel$_c$ | 83.06±0.73 | 27.84±0.59 | 20.77±0.47 | 18.75±0.64 | 14.33±0.71 |
| SAAT-minimax$_h$ | 85.29±0.68 | 61.52±0.85 | **53.86±1.37** | 47.00±0.96 | 49.84±0.33 |
| SAAT-minimax$_c$ | **86.98±0.52** | **63.73±0.72** | 51.70±0.66 | **49.37±0.84** | **50.68±0.54** |

Table 4: Test accuracy (%) of various defense methods trained on CIFAR100 with $l_\infty$-norm constrained perturbations on DNNs. (The results of Natural on clean data are just baselines for reference.)

| Method | Clean | FGSM | 10-PGD | CW | AutoAttack |
|---|---|---|---|---|---|
| Natural | *77.79±0.47* | 1.16±0.21 | 0.00±0.00 | 0.00±0.00 | 0.00±0.00 |
| Standard | 58.13±0.36 | 35.99±0.44 | 27.20±0.72 | 23.64±0.77 | 22.63±0.94 |
| IAAT | 59.47±0.37 | 27.23±0.65 | 18.58±0.59 | 17.79±0.34 | 13.73±0.88 |
| MMA | 49.03±0.77 | 27.79±0.74 | 20.50±0.67 | 18.96±0.51 | 15.39±0.74 |
| FAT | 61.28±0.88 | 25.03±0.72 | 19.73±0.56 | 19.92±0.81 | 13.09±0.69 |
| TRADES | 50.71±0.67 | 28.99±0.84 | 21.59±0.65 | 17.41±0.57 | 16.47±0.77 |
| SAAT-kernel$_h$ | 66.88±0.73 | 13.64±0.81 | 8.77±0.61 | 7.65±0.84 | 7.98±0.72 |
| SAAT-kernel$_c$ | 68.56±0.53 | 12.71±0.66 | 6.79±0.75 | 6.87±0.52 | 5.34±0.69 |
| SAAT-minimax$_h$ | **70.72±0.47** | **47.18±0.33** | **39.68±0.51** | **34.70±0.36** | **32.77±0.57** |
| SAAT-minimax$_c$ | 68.11±0.57 | 43.80±0.44 | 35.33±0.39 | 31.69±0.67 | 30.83±0.62 |

### 5.1.2 ATTACK SETTINGS:

Four popular attack methods are used in the experiments: FGSM (Goodfellow et al., 2014), 10-PGD (PGD with 10 steps) (Madry et al., 2017), CW (Carlini & Wagner, 2017) and AutoAttack (Croce & Hein, 2020). All the attacks can be performed with both $l_2$ and $l_\infty$ versions. In the $l_\infty$ version,

for FGSM and 10-PGD, the perturbation radius is set as $\epsilon_{test} = 0.3$ for MNIST and $\epsilon_{test} = 8/255$ for CIFAR10 and CIFAR100, the step size for 10-PGD is $\epsilon_{test}/4$, which is a standard setting for adversarial attack (Madry et al., 2017; Ding et al., 2018).

### 5.1.3 Implementation Details:

Under $l_\infty$-norm constrained perturbations, we set $\epsilon_{\max} = 0.3$ for MNIST, $\epsilon_{\max} = 8/255$ for CIFAR10 and Tiny Imagenet, and the step size is set as $\epsilon_{\max}/4$. For all algorithms, we set the batch size as 100 with 10 epochs. We use 5-fold cross validation to choose the optimal learning rate $\gamma \in 2^{[-3,3]}$.

We use the PreAct ResNet18 architecture for CIFAR10 and CIFAR100 and use two convolutional networks with 16 and 32 convolutional filters followed by a fully connected layer of 100 units for MNIST, which are the same model structures provided by Wong et al. (2020). For all the compared algorithms, we use the cross entropy loss function. For SAAT-minimax and SAAT-kernel, $b_i$ is gotten by $\hat{l}(x_i, y_i, f_w, \epsilon_{\max}) - k_i \epsilon_{\max}$, we set $k_i = 2$, $\eta = 0.05$, linearly increase regularization parameter $\lambda$ from 1 to 3 on MNIST, set $k_i = 0.15$, $\eta = 0.3/255$, linearly increase $\lambda$ from 0 to 0.5 on CIFAR10, and set $k_i = 0.3$, $\eta = 0.3/255$, linearly increase $\lambda$ from 0 to 0.6 on CIFAR100.

### 5.2 Experimental Results and Analyses

**Robustness against various attacks.** We first explore robustness of adversarial training algorithms against different attacks in Tables 2, 3, 4. It can be seen clearly that our SAAT-minimax not only improves natural generalization performance, but also enjoys stronger defensive ability against various adversarial examples. Moreover, it indicates that hinge loss contributes to higher adversarial robustness than cross entropy. Although SAAT-kernel is not as robustness as adversarial training algorithms on the minimax problem, it largely improves accuracy on clean data. As for the compared algorithms, although they improve the generalization performance on clean data to some extent, they sacrifice much robustness on strong attacks, especially CW and AutoAttack.

**Running time with different sizes of training samples.** Fig. 3 shows the running time of various adversarial training algorithms when training samples of different sizes. We can find that SAAT-kernel is much more efficient due to its one-layer objective function. For other algorithms on the minimax problem, the time-consuming factor lies on the $K$-step PGD attack. Among adversarial training algorithms with adaptive $\epsilon_i$, SAAT-minimax is superior to others since it avoids brute-force search for the optimal $\epsilon_i^*$. We also note that the time of SAAT-minimax costs a little longer than that of FAT since FAT applies the early-stopped PGD strategy. But the extra time can be ignored compared with the superiority we have in robustness and generalization.

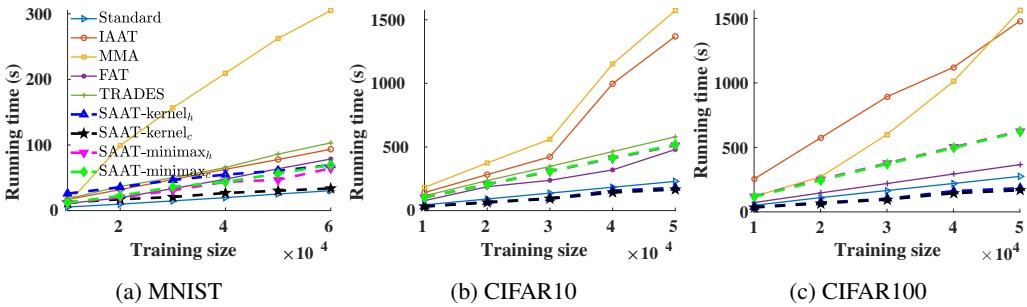

(a) MNIST        (b) CIFAR10        (c) CIFAR100

Figure 3: Running time of adversarial training algorithms against different sizes of training samples.

## 6 Conclusion

To achieve a better trade-off between robustness and accuracy without much computation overhead, in this paper, we propose an adversarial training framework with self-adaptive perturbation radii named SAAT. This framework can also get the closed-form solution of the optimal perturbation radii and avoids tedious searching compared with existing works, which is applicable to both DNNs and kernel SVMs. Comprehensive experimental results verify that our algorithms not only improve adversarial robustness and natural generalization, but also can be competitive with other adversarial training algorithms in terms of running time.

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
