# OpenReview forum: "Self-Adaptive Perturbation Radii for Adversarial Training"
_ICLR.cc/2023/Conference — Submitted to ICLR 2023_

### Official Review · Reviewer_CG7K · 2022-10-25

**Confidence:** 4
**Correctness:** 3
**Technical Novelty And Significance:** 2
**Empirical Novelty And Significance:** 3
**Recommendation:** 3

**Clarity, Quality, Novelty And Reproducibility:**

Clarity is good.

Quality is fair.

Novelty is fair.

Reproducibility is good.

**Strength And Weaknesses:**

Strength: This paper is working on a meaningful problem. Finding adaptive epsilon per training sample could be the key to break the accuracy-robustness trade-off and do well on both clean and perturbed data.

Weaknesses:

1. The proposed heuristics involve substantial approximation.
First of all, the proposed loss function (5) is already a heuristic, -- it's assuming that the optimal epsilon_i is such that the derivative of perturbed-loss with respect to epsilon_i is equal to lambda. There is no guarantee that the derivative of perturbed-loss with respect to epsilon is a good metric to use to determine optimal epsilon.
On top of that, the proposal is not to solve (5) directly but rather to use a crude heuristic described in section 3.3, equation (7) and the paragraph after it. There are more heuristics in section 4 but experimental results suggest that they are worse than that in section 3.3.
With such approximation on top of approximation, it's unclear whether the final algorithm is a meaningful solution to the original problem.

2. In the main results tables 2 and 3, the "Standard" rows are substantially worse than the numbers reported in (Madry et al., 2017). Please explain. If compared with the original numbers in (Madry et al., 2017), it seems that the proposed method is only marginally better. Also for TRADES, tables 2 and 3 suggest that the proposed method is only marginally better. Results in table 4 are strong and it's worth adding detailed analysis on what makes CIFAR100 more favorable for the proposed method.


**Summary Of The Paper:**

This paper proposes a method to perform adversarial training with adaptive epsilon per training sample. Several heuristics are proposed to determine epsilon per sample. The benefit is improved accuracy-robustness trade-off.

**Summary Of The Review:**

This paper is addressing a meaningful problem. However, the proposed method is a crude heuristic and may need improvement, and there are some questionable numbers in the experimental results.

---

### Official Review · Reviewer_qfVT · 2022-10-25

**Confidence:** 4
**Clarity, Quality, Novelty And Reproducibility:** The novelty of this paper is limited …
**Correctness:** 2
**Technical Novelty And Significance:** 2
**Empirical Novelty And Significance:** 2
**Recommendation:** 5

**Strength And Weaknesses:**

How to find a suitable epsilon is a widely studied problem, including some methods mentioned in the paper, and [1]. [1] tested the effect of different epsilon variation strategies on robustness, including Constant, Linear, Cosine, and their version of period reset. This paper also adopts a linear schedule, hope the authors discuss and compare their method with [1].
The main mechanism of the proposed method is "Specifically, if the PGD attack fails to find an adversarial image xi + δi that can be misclassified, it implies ε∗i is too small. Thus, we set ε∗i = ε∗i + η. Otherwise, we set ε∗i = ε∗i − η, where η is a pre-specified fixed step size." Actually, the method is not novel from my perspective, and it provides little insight.

Some questions about the experiments:
1. The result of the AA attack against AT and TRADES in Table 4 seems too low, some papers have tested under similar settings, and performs better results [2, 3]. Could the authors have a little check?
2. What about the performance of TRADES + SAAT?
3. How the robustness will increase when training on a larger model, e.g., WRN-34-10? And what about the effective improvement in robustness when training with additional data?


[1] Liu C, et al. On the loss landscape of adversarial training: Identifying challenges and how to overcome them. In NeurIps 2020.

[2] Dong Y, et al. Exploring Memorization in Adversarial Training. In ICLR 2022.

[3] Wu D, et al. Adversarial weight perturbation helps robust generalization. In NeurIps 2020.

**Summary Of The Paper:**

The paper advocate assigning adaptive perturbation constraint $\epsilon$ to adversarial examples, during the inner maximization in adversarial training. Experiment results show that the SAAT method can improve the adversarial robustness compared with several existing methods.

**Summary Of The Review:**

The contribution is limited, and the empirical comparisons are not convincing to me.

---

### Official Review · Reviewer_LAk2 · 2022-10-25

**Confidence:** 5
**Clarity, Quality, Novelty And Reproducibility:** See the above.
**Correctness:** 2
**Technical Novelty And Significance:** 2
**Empirical Novelty And Significance:** 2
**Recommendation:** 6

**Strength And Weaknesses:**

Strength
- Well written.
- Propose an efficient approach for searching for adaptive perturbation radii for each data point. It has better performance thant the compared methods.

Weakness
- Regarding the robustness performance, the compared methods are far from SOTA. The current robustenss performance of the proposed approach is not significant.
- It is claimed that the adaptive radii could help allievate the accuracy-robustness trade-off. However, according to the experimental results,  the natural accuracy does not change much except the cifar100 dataset. Could you explain why the case in this dataset?

**Summary Of The Paper:**

This paper proposes a new method for searching for optimal perturbation radii for adversarial training, which is shown to be more efficient than existing works.

**Summary Of The Review:**

Personally I appreciate the approach proposed by the authors, but the performance is not that significant. Thus I suggest weak acceptance.

---

### Decision · Program_Chairs · 2023-01-20

**Decision:**

Reject

**Justification For Why Not Higher Score:**

Valid questions of the reviewers are not answered by the authors.

**Justification For Why Not Lower Score:**

N/A

**Metareview: Summary, Strengths And Weaknesses:**

Two reviewers suggest rejection, one reviewer suggests weak acceptance. All reviewers raised questions about the experimental results, the employed assumptions and other issues. The authors did not answer these questions in the rebuttal and as I think that these questions are valid, I have to reject the paper.